# Temporal-Aware Iterative Speech Model for Dementia Detection

## Abstract

Current acoustic markers for dementia detection often rely on static feature aggregation or error-prone linguistic transcription (ASR), thereby failing to capture the fine-grained, frame-to-frame temporal deterioration of articulatory motor control. To address this, we introduce TAI-Speech, an ASR-free framework that models speech deterioration as a continuous temporal trajectory analogous to physical motion. Our architecture introduces two key innovations: 1) Optical Flow-inspired Iterative Refinement: By treating spectrograms as sequential frames, this component uses a convolutional GRU to capture the fine-grained, frame-to-frame evolution of acoustic features; and 2) Cross-Modal Attention, which dynamically aligns spectral features with prosodic contours (pitch and pauses) to detect pathological mismatches. Experimental evaluation on the DementiaBank Corpus demonstrates that TAI-Speech achieves an AUC of 83.9% and Recall of 89.0%. Importantly, our model surpasses strong state-of-the-art baselines on AUC ROC, including fine-tuned Wav2Vec 2.0 (67.9%), Audio Spectrogram Transformers (74.8%), and CNNs (76.8%). These results confirm that explicitly modeling acoustic flow yields superior diagnostic sensitivity compared to latent linguistic representations or static classifiers, offering a privacy-preserving and computationally efficient solution for early cognitive screening.

## 1 Introduction

Dementia is a progressive neurodegenerative syndrome currently affecting an estimated 55 million people worldwide, with prevalence projected to rise sharply by 2050. It is marked by gradual decline in memory, language, and executive function, and Alzheimer's disease remains the most common subtype (Ortiz-Perez et al. (2023),Pan et al. (2025),Agbavor & Liang (2022),Galanakis et al. (2025)). Early detection is critical for timely intervention and improved quality of life (Ortiz-Perez et al. (2023),Gkoumas et al. (2024)). Among the most promising non-invasive biomarkers are speech and language changes, which often appear during preclinical stages (Gkoumas et al. (2024),Agbavor & Liang (2022),Pan et al. (2025),Li et al. (2025),Kannojia et al. (2025),Yeung et al. (2021)).

Speech deterioration is closely tied to functional decline measured by Instrumental Activities of Daily Living (IADLs) abilities such as financial management, medication adherence, and complex communication (Fieo et al. (2014),Laurentiev et al. (2024),Fieo & Stern (2018)). Extended IADL (x-IADL) scales correlate strongly with language function, processing speed, and visuospatial ability (Fieo et al. (2014)). Despite extensive work analyzing speech or IADLs separately, current methods rarely model their temporal interdependence, even though language decline, commonly characterized as slowed speech, lexical retrieval failures, and reduced syntactic complexity, often precedes measurable IADL impairment (Yeung et al. (2021),Chen & Li (2024)).

We hypothesize that gradual, fine-grained deterioration of speech is a precursor to IADL impairment and can be captured by an architecture inspired by optical-flow estimation. Both problems require tracking continuous temporal changes via correspondence analysis and iterative refinement (Alfarano et al. (2024),Teed & Deng (2021)). Analogous to how optical flow estimates motion between video frames, our approach models the temporal evolution of spectrogram frames, allowing precise characterization of pauses, pitch variability, and other subtle acoustic patterns.

We present TAI-Speech, a deep learning framework that treats speech not as a static signal, but as a dynamic sequence of evolving spectrogram frames. Based on cross-modal research demonstrating

that physical articulatory motions and acoustic signals share a consistent temporal and structural correspondence (Zhao et al. (2018),Ephrat et al. (2018)), we adapt the Recurrent All-Pairs Field Transform (RAFT) paradigm (Alfarano et al. (2024),Teed & Deng (2021),Sui et al. (2022)) to audio analysis. Rather than estimating explicit motion vectors, we leverage RAFT's iterative refinement to construct a temporally aware embedding that captures the "velocity" of spectral degradation. A convolutional GRU serves as the recurrent update module, iteratively refining these latent representations, while cross-attention dynamically aligns acoustic features with prosodic cues. While our conceptual framework is motivated by the functional deterioration seen in IADLs, we empirically validate TAI-Speech as a detector of acoustic motor instability the upstream mechanistic failure that serves as a proximal biomarker for downstream functional decline. Evaluated on the DementiaBank corpus, our approach outperforms strong linguistic baselines, demonstrating that modeling these temporal dynamics offers a robust, ASR-free alternative for early detection.

**Our Contribution**

- We adapted iterative-refinement priors from optical-flow modeling (Teed & Deng, 2021) to audio by treating non-stationary acoustic variation as continuous manifold evolution.
- Propose a dual-stream ASR-free architecture with a temporal-consistency objective and cross-modal attention linking spectral and prosodic dynamics.
- Establish a temporally regularized training framework that enforces physically grounded continuity in acoustic trajectories, enabling strong performance in low-resource clinical-speech settings without transcription.

## 2 RELATED WORK

### 2.1 COMPUTATIONAL APPROACHES TO SPEECH BASED DEMENTIA DETECTION

Speech analysis has emerged as a non-invasive, cost-effective modality for early dementia diagnosis and monitoring (Ortiz-Perez et al. (2023); Agbavor & Liang (2022); Braun et al. (2024)). A cornerstone resource is the DementiaBank Corpus, which records subjects describing the "Cookie Theft" picture see Figure 1 to elicit lexical retrieval challenges and discourse impairments. Its derivatives, ADReSS and ADReSSo provide balanced demographics and higher acoustic quality, supporting tasks such as Alzheimer's disease classification, MMSE regression, and cognitive-decline prediction, with ADReSSo emphasizing speech-only input and ASR-generated transcripts (Luz et al. (2021)).

Feature extraction spans acoustic (log-Mel spectrograms, MFCCs, energy contours, pauses, hesitations) and linguistic (vocabulary richness, syntactic complexity, POS distributions, disfluency metrics) domains (Ortiz-Perez et al. (2023); Braun et al. (2024); Ilias & Askounis (2023); Woszczyk et al. (2024)). Deep models dominate: CNNs for audio, RNN/LSTM and Transformer variants (e.g., BERT, RoBERTa, DeiT, GPT-3) for both acoustic and text representations (Pan et al. (2025); Braun et al. (2024); Meilán et al. (2014); König et al. (2018); Gong et al. (2021a)). Self-supervised models such as wav2vec 2.0 capture rich acoustic embeddings with strong downstream performance (Pan et al. (2025); Braun et al. (2024)).

Multimodal fusion strategies integrate modalities through early feature concatenation, late decision-level aggregation, and cross-attention mechanisms that dynamically weight each modality. Although ASR errors can introduce noise, transcripts with relatively high Word Error Rates (WER) often perform on par with or better than manual transcriptions for dementia classification, suggesting that salient cognitive cues persist in noisy outputs (Pan et al. (2025); Shon et al. (2023)).

### 2.2 CORRELATING SPEECH WITH FUNCTIONAL DECLINE

Loss of independence in Instrumental Activities of Daily Living (IADLs) is a defining clinical marker of dementia (Fieo et al. (2014); Liepelt-Scarfone et al. (2013)). Modern, technology-mediated IADLs such as online financial tasks or text messaging—offer even greater sensitivity for early Alzheimer's detection (Benge et al. (2024)). Numerous studies report strong links between speech abilities and functional status: language deficits often precede measurable IADL impairment ( Gkoumas et al. (2024); Yeung et al. (2021)).

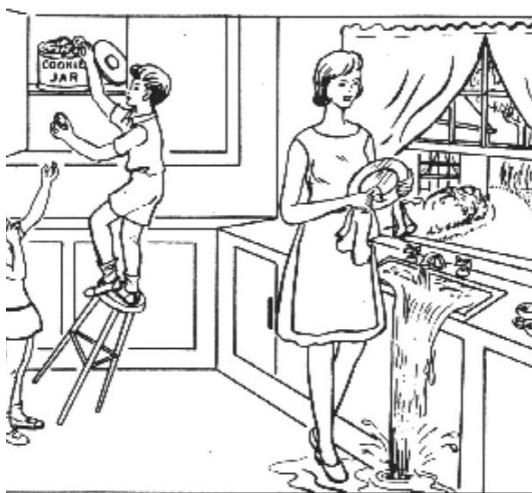

Figure 1: Cookie Theft picture

Automated speech and language analysis captures these associations objectively. Word-finding difficulty correlates with increased pause frequency and specific acoustic signatures (e.g., MFCC patterns), while incoherence and perseveration manifest as degraded discourse structure and repeated utterances measurable via cosine similarity. Reduced lexical diversity, simplified grammar, and malformed verb phrases signal syntactic and semantic breakdown (Yeung et al. (2021)). Beyond structured tasks, NLP methods applied to unstructured clinical narratives in EHRs extract indicators of IADL/ADL impairment, enabling scalable integration of functional status into research and clinical decision support (Laurentiev et al. (2024); Penfold et al. (2022)).

### 2.3 Iterative Refinement in Optical Flow

Optical flow research has advanced from classical variational formulations to deep iterative refinement. RAFT achieves state-of-the-art accuracy by maintaining high-resolution flow fields and leveraging multi-scale correlation volumes with a convolutional GRU for recurrent updates ( Teed & Deng (2021); Sui et al. (2022)). Follow-on architectures such as IRR and LiteFlowNet refine cost volumes through shared-weight cascades and optimized objectives like RCELoss (Alfarano et al. (2024); Hui et al. (2018)). These iterative principles, high-resolution correlation, recurrent updates, and context preservation inform broader temporal modeling strategies and motivate cross-domain applications beyond computer vision ( Alfarano et al. (2024); Meßmer et al. (2025)).

### 2.4 Theoretical Frame for Temporal Analysis

Speech is a fundamentally temporal information modality, where the state at a given moment is intrinsically linked to its context. Temporal aspects in audio, such as hesitation and pauses, speaking rate, and word duration, serve as significant indicators of cognitive decline Xu et al. (2023). Dementia recordings exhibit prolonged utterances and characteristic pause patterns, with manual transcripts often marking these events explicitly (Ortiz-Perez et al. (2023),Pan et al. (2025),Braun et al. (2024)). Acoustic representations such as log-Mel spectrograms, MFCCs, and eGeMAPS capture short-term spectral and physiological voice dynamics (GOrtiz-Perez et al. (2023),Gong et al. (2021a),Corvitto et al. (2024), Luz et al. (2021)).

Longitudinal analysis tracks language change across sessions via embedding similarity and related metrics ( Gkoumas et al. (2024),Braun et al. (2024)). Linguistic deficits, empty speech, circumlocution, repetition, poor grammar are temporal manifestations of cognitive decline (Chen & Li (2024)). Extra-linguistic cues such as keystroke pauses in written text further complement audio evidence (Gkoumas et al. (2024)).

Context-aware large language models can exploit preceding audio or text to predict next-sentence semantics or topic flow, enhancing downstream temporal reasoning (Shon et al. (2023),Bai et al. (2024)). Related techniques in audio-visual segmentation similarly rely on temporal consistency, where optical flow provides low-level motion signals for tasks like emotion recognition and lip-reading (Alfarano et al. (2024),Torabi & Nilchi (2014)). Temporal Enhancement Modules (TEM) extend these ideas by exchanging learnable context tokens across frames to strengthen inter-frame coherence (Geng & Gu (2025)).

## 2.5 CROSS-MODAL DYNAMIC COUPLING

Foundational work in The Sound of Pixels (Zhao et al. (2018)), sound of motion (Zhao et al. (2019)) and visual speech recognition (Ephrat et al. (2018)) establishes that acoustic output is the causal result of physical motor dynamics. These studies demonstrate that while the mapping is not strictly bijective, the temporal derivatives of physical motion (e.g., lip velocity) are structurally preserved in the evolution of the acoustic manifold. This kinematic-acoustic coupling provides the theoretical justification for applying motion-tracking architectures specifically Optical Flow priors to purely acoustic time-series, as the velocity of spectral degradation serves as a proxy for the underlying articulatory drift.

## 3 METHODOLOGY

### 3.1 TASK AND DATASET

We evaluate our approach on spontaneous picture description, a standardized neurocognitive paradigm used to probe semantic memory and episodic retrieval (Mueller et al. (2018)). Participants describe the Cookie Theft line drawing (Lanzi et al. (2023)), producing naturalistic speech that reveals lexical retrieval difficulty, hesitations, and discourse-level impairments.

Experiments use the DementiaBank Corpus, the largest publicly available speech dataset for cognitive-impairment assessment. We focus on the clinically validated subsets comprising 222 recordings from 89 healthy controls (HC) and 255 recordings from 168 participants with Alzheimer's disease (AD), for a total of 477 audio samples. All recordings are accompanied by diagnostic annotations and are sampled at 16 kHz.

### 3.2 MODEL ARCHITECTURE

Our goal is to capture temporal markers of functional decline, particularly those linked to Instrumental Activities of Daily Living directly from raw speech. **TAI-Speech** integrates prosodic encodings, convolutional spectral processing, iterative temporal refinement, and sequence-level aggregation (Figure 2). The model is trained end-to-end with a joint objective combining cross-entropy classification and a temporal smoothness regularizer to enforce stability across successive frames.

### 3.3 FEATURE ENCODINGS

Raw speech $x(t)$ is first resampled and transformed using the short-time Fourier transform (STFT). A log-Mel spectrogram is computed:

$$S(m,n) = \log\Big(\sum_k |X(k,n)|^2 H_m(k)\Big), \qquad (1)$$

where $X(k,n)$ is the STFT coefficient at frequency $k$ and frame $n$, and $H_m$ is the $m$-th Mel filter. Tabler 1 summaries the notation

Prosodic correlates relevant to IADL are explicitly extracted: (i) normalized pitch track $\tilde{p}(n)$, and (ii) pause probability $q(n)$ estimated from voice activity detection. These auxiliary encodings are fused into a joint representation:

$$z(n) = \phi\big(W_f[\tilde{p}(n), q(n)] + b_f\big), \qquad (2)$$

where $W_f$ and $b_f$ are trainable parameters and $\phi(\cdot)$ is a non-linear activation.

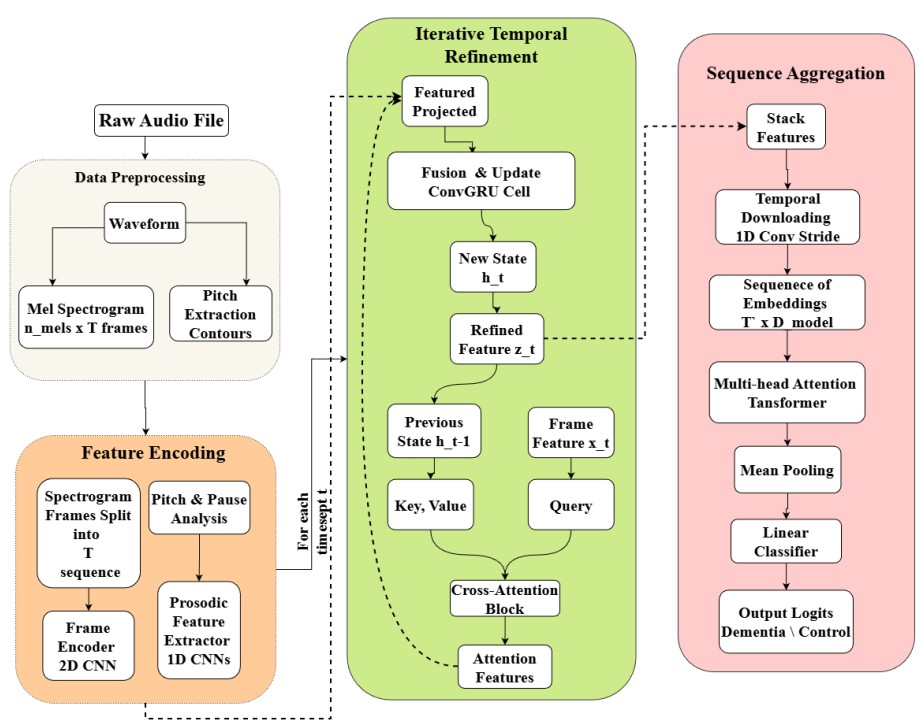

Figure 2: Architecture Model

Table 1: Summary of Notations and Their Meanings

| Symbol | Meaning | Symbol | Meaning |
|---|---|---|---|
| $x(t)$ | Raw speech waveform | $S(m, n)$ | Log-Mel spectrogram value |
| $X(k, n)$ | STFT coefficient | $H_m(k)$ | $m$-th Mel filter |
| $\tilde{p}(n)$ | Normalized pitch track | $q(n)$ | Pause probability |
| $z(n)$ | Prosodic feature vector | $W_f, b_f$ | Trainable weight and bias |
| $\phi(\cdot)$ | Non-linear activation | $h^{(l)}$ | Local spectral embedding |
| Attn(Q,K,V) | Cross-modal attention | $Q, K, V$ | Query, Key, Value matrices |
| $r_t, u_t$ | Reset and update gates | $H_t$ | Hidden state at time $t$ |
| $\tilde{H}_t$ | Candidate hidden state | $\odot$ | Element-wise multiplication |
| $*$ | Convolution operation | $\{h_1, \ldots, h_T\}$ | Sequence of embeddings |
| $u_{\text{cls}}$ | Classification token | $U$ | Transformer output |
| $\hat{y}$ | Predicted probability | $W_c, b_c$ | Classification layer params |
| $L$ | Total training loss | $L_{CE}$ | Cross-entropy loss |
| $\lambda_{\text{cls}}, \lambda_{\text{temp}}$ | Regularization weights | $\|h_t - h_{t-1}\|_2^2$ | Temporal consistency term |

### 3.4 TEMPORAL REFINEMENT MODULES

#### 3.4.1 CROSS-ATTENTION CONTEXTUALIZATION

The spectral encoder produces local embeddings $h^{(l)}$. To integrate prosodic factors, a cross-modal attention module computes:

$$\text{Attn}(Q, K, V) = \text{softmax}\Big(\frac{QK^\top}{\sqrt{d}}\Big)V, \tag{3}$$

where queries $Q$ are derived from spectro-temporal features, while keys and values come from $z(n)$. This aligns acoustic features with temporal dynamics of pitch and pause, providing contextualized embeddings.

#### 3.4.2 ITERATIVE UPDATE BLOCK

To refine temporal representations, we employ a multi-scale ConvGRU. At each time step $t$, the hidden state $H_t$ is updated by:

$$r_t = \sigma(W_r * x_t + U_r * H_{t-1}), \tag{4}$$
$$u_t = \sigma(W_u * x_t + U_u * H_{t-1}), \tag{5}$$
$$\tilde{H}_t = \tanh(W * x_t + U * (r_t \odot H_{t-1})), \tag{6}$$
$$H_t = u_t \odot H_{t-1} + (1 - u_t) \odot \tilde{H}_t, \tag{7}$$

where $*$ denotes convolution and $\odot$ elementwise multiplication. This iterative block progressively corrects and stabilizes features across multiple scales, reflecting temporal organization in speech.

### 3.5 SEQUENCE AGGREGATION AND CLASSIFICATION

Downsampled embeddings $\{h_1, \ldots, h_T\}$ are passed into a Transformer encoder augmented with a classification token $u_{\text{cls}}$. The self-attention mechanism models higher-order dependencies:

$$U = \text{Transformer}([u_{\text{cls}}, h_1, \ldots, h_T]). \tag{8}$$

The final classification is computed as:

$$\hat{y} = \text{softmax}(W_c u'_{\text{cls}} + b_c), \tag{9}$$

where $u'_{\text{cls}}$ is the contextualized embedding.

The training objective combines cross-entropy loss with a temporal consistency regularizer:

$$\mathcal{L} = \lambda_{\text{cls}}\mathcal{L}_{\text{CE}}(\hat{y}, y) + \lambda_{\text{temp}}\frac{1}{T-1}\sum_{t=2}^{T}\|h_t - h_{t-1}\|_2^2, \tag{10}$$

encouraging stability in temporal encodings while preserving discriminative capacity. While the first term minimize standard cross-entropy error for classification, the second term temporal consistency regularizer, penalizes the squared Euclidean distance between consecutive hidden states $(h_t, h_{t-1})$ to enforce smooth latent evolution. This constraint ensures that learned features reflect the continuous acoustic drift of physiological decline rather than overfitting to transient noise artifacts.

## 4 EXPERIMENTAL SETUP

### 4.1 EVALUATION PROTOCOL

In order to guarantee a rigorous and unbiased evaluation of the proposed approach, we adopt a stratified five-fold cross-validation (5-fold CV) protocol. This strategy preserves the original class distribution within each fold, a critical consideration when working with imbalanced clinical datasets. The primary evaluation metric is the Area Under the Curve (AUC), which provides a robust measure of discriminative capability between dementia and healthy control groups. In addition, we report secondary performance indicators, namely accuracy, precision, recall, and F1-score, thereby offering a comprehensive assessment across multiple dimensions of classification performance.

---

**Algorithm 1:** TAI-Speech: Temporal–Acoustic–IADL Speech Classification

---

**Input:** Raw waveform $x(t)$,;
ground-truth label $y$
**Output:** Predicted probability $\hat{y}$
```
# Preprocessing
```
Resample $x(t)$ and compute log-Mel spectrogram $S(m,n) = \log \sum_k |X(k,n)|^2 H_m(k)$
Extract normalized pitch $\tilde{p}(n)$ and pause probability $q(n)$
Fuse prosodic vector $z(n) = \phi(W_f[\tilde{p}(n), q(n)] + b_f)$
```
# Spectral Encoding
```
$h(l) \leftarrow$ Hierarchical convolutional encoder on $S(m,n)$
```
# Cross-Attention Contextualization
```
$h'(l) \leftarrow \text{Attn}(Q, K, V)$ with $Q = h(l)$,;
$K, V = z(n)$
```
# Iterative Temporal Refinement
```
**for** $t = 1$ **to** $T$ **do**
$\quad r_t = \sigma(W_r * x_t + U_r * H_{t-1})$
$\quad u_t = \sigma(W_u * x_t + U_u * H_{t-1})$
$\quad \tilde{H}_t = \tanh(W * x_t + U * (r_t \odot H_{t-1}))$
$\quad H_t = u_t \odot H_{t-1} + (1 - u_t) \odot \tilde{H}_t$
```
# Sequence Aggregation and Classification
```
$U \leftarrow \text{Transformer}([u_{\text{cls}}, h'_1, \ldots, h'_T])$
$\hat{y} \leftarrow \text{softmax}(W_c u'_{\text{cls}} + b_c)$
```
# Training Loss
```
$\mathcal{L} = \lambda_{\text{cls}} \mathcal{L}_{\text{CE}}(\hat{y}, y) + \lambda_{\text{temp}} \frac{1}{T-1} \sum_{t=2}^{T} \|h_t - h_{t-1}\|_2^2$
**return** $\hat{y}$

---

## 4.2 BASELINE SYSTEMS

To rigorously evaluate the proposed temporal modeling approach, we benchmark TAI-Speech against three distinct architectural paradigms trained on the DementiaBank Corpus:

- Self-Supervised Foundation (Wav2Vec 2.0): We fine-tuned Wav2Vec 2.0 Baevski et al. (2020) to evaluate the efficacy of latent linguistic representations pre-trained on healthy speech for detecting pathological acoustic drift.

- Transformer-based (AST): We implemented the Audio Spectrogram Transformer Gong et al. (2021b), which utilizes patch-based global self-attention, to assess whether global context modeling suffices compared to frame-by-frame recurrent refinement.

- Static Convolutional (2D CNN): The 2D CNN Hershey et al. (2017) served as a time-agnostic baseline to isolate the contribution of temporal dynamics versus static feature aggregation.

Additionally, we reference recent multimodal state-of-the-art systems Braun et al. (2024); Pan et al. (2025) to demonstrate the comparative efficacy of our ASR-free approach against text-dependent architectures.

## 4.3 PROPOSED SYSTEM

**Algorithm.** Algorithm 1 presents the overall procedure of our proposed method. The TAI-Speech framework refines acoustic representations of spontaneous speech to detect dementia-related functional decline. The procedure can be summarized in three stages:

- **Acoustic Feature Encoding:** Raw audio $x(t)$ is converted into log-Mel spectrogram frames $S(m,n)$. A hierarchical convolutional encoder extracts local spectral representations as initial feature maps.

- **Iterative Temporal Refinement:** Hidden states $H_t$ are updated with a multi-scale ConvGRU to capture long-range temporal context. The prosodic characteristics, the normalized pitch $\tilde{p}(n)$ and the probability of pause $q(n)$, are fused using a cross-modal attention layer for richer temporal contextualization.

- **Sequence Aggregation and Classification:** Refined embeddings are downsampled and passed through a Transformer encoder with a learnable classification token $u_{\text{cls}}$. A final linear layer with softmax outputs the dementia vs. control prediction. Training employs a cross-entropy loss plus a temporal-smoothness regularizer to encourage frame-to-frame consistency.

### 4.4 Training Details

**Signal Processing and Feature Extraction.**   Raw audio is resampled to 16 kHz. Log-Mel spectrograms are computed using an STFT with a window size ($n\_fft$) of 2048, a hop length of 512, and 64 Mel filterbanks spanning 0–8000 Hz. Prosodic features are extracted using `librosa.piptrack`, with pitch constrained to 75–400 Hz. Voice activity detection is performed using an energy-based method with 25 ms frames and a normalized energy threshold of 0.01 to generate pause-probability sequences.

**Training Configuration.**   All models are trained under a unified protocol for comparability. Optimization uses AdamW with a batch size of 4, a maximum of 200 epochs, and an initial learning rate of $1 \times 10^{-5}$. Early stopping with a patience of 10 epochs based on validation AUC is applied to mitigate overfitting. Class imbalance is handled via a `WeightedRandomSampler` to maintain uniform class representation per batch. Training is conducted on a single NVIDIA RTX A4000 GPU (32 GB).

## 5 Results and Discussion

This section presents the performance of our proposed architecture, contextualizes the findings by comparing them against established baseline models, and discusses the broader implications of our results, with a specific focus on how the model's design relates to the detection of functional decline.

### 5.1 Quantitative Performance Analysis

The proposed model was rigorously evaluated using a 5-fold cross-validation protocol. The primary metric for assessing the model's ability to discriminate between dementia and healthy control classes was the Area Under the Curve (AUC), with Accuracy (ACC), Recall (REC), and F1-score also reported for a comprehensive analysis.

As summarized in Table 2, our proposed architecture achieved a high level of discriminative performance, yielding an test AUC of 0.839. The model obtained an accuracy of 80.55%, recall of 0.890, and an F1-score of 0.813.

Table 2: The Result (%) of our Model

| System | AUC | ACC | REC | F1-score |
|---|---|---|---|---|
| Our Model | 83.9 | 0.81 | 0.890 | 0.813 |

For evaluation, we benchmarked our system against previously reported baseline models, as well as Transformer-based acoustic approaches that analyze transcribed speech.

The TAI-Speech architecture achieved an AUC of 0.839, an accuracy of 80.55%, a recall of 89.0%, and an F1-score of 0.813. These results represent a significant improvement over purely linguistic baselines. When benchmarked against state-of-the-art systems in table 3, our model demonstrates good performance across all evaluation metrics. The 8% improvement in AUC over Braun et al. (2024).'s pause-infused text model (77.2%) and competitive performance against Pan et al. (2025) attention-based multimodal system (82.56% accuracy) underscore the efficacy of our temporatsively on acoustic signals without requiring error-prone ASR transcription or linguistic feature extraction.

Table 3: Performance comparison between the proposed model with other modalities

| System | Modality | AUC (%) | Acc (%) | Recall | F1-score |
|---|---|---|---|---|---|
| Pan et al. (2019) | Lingustic | – | 70.83 | 0.71 | 0.70 |
| Pan et al. (2025) | Multimodal | – | **82.56** | 0.83 | **0.83** |
| Braun et al. (2024) | Multimodal | 77.2 | – | – | – |
| Our Model | Acoustic-Temporal | 83.9 | 80.55 | **0.89** | 0.83 |

While systems incorporating ASR features achieve the highest AUC and accuracy, our purely acoustic model obtains the highest recall. It is notable that TAI-Speech achieves this level of performance without relying on a linguistic pipeline. This suggests that the temporal dynamics encoded within the acoustic signal contain sufficient information for effective dementia classification. This single-modality approach may offer advantages in robustness and simplicity, as it avoids potential cascading errors from ASR systems, which can struggle with the atypical speech patterns often present in clinical populations. The results indicate that direct modeling of speech dynamics is a viable and powerful alternative to multimodal approaches that require transcription.

**Quantitative Analysis of Baselines.** As shown in Table 4, TAI-SPEECH achieves the best overall performance (AUC 0.83, Recall 0.89), validating the value of temporally regularized acoustic modeling. Foundation models like Wav2Vec 2.0 underperform (AUC 0.679, Acc 0.565), likely due to pre-training on healthy speech that biases toward linguistic clarity, filtering out pathological drift signals critical in dementia. AST (AUC 0.748) highlights the limitations of patch-based attention: global spectrogram partitioning disrupts the continuity needed to track motor decline. ResNet50 (AUC 0.768) performs better by capturing local spectral texture. The 6.2% AUC gain from TAI-Speech over ResNet confirms the necessity of modeling acoustic flow frame-to-frame evolution as opposed to static aggregation. Clinically, the high Recall (0.89) positions TAI-Speech as a reliable screening tool, minimizing false negatives in early-stage detection.

Table 4: Performance comparison between the proposed model with baseline models

| System | AUC (%) | Acc (%) | Recall | F1-score |
|---|---|---|---|---|
| AST | 0.7481 | 0.698 | 0.698 | 0.705 |
| Wav2vec 2.0 | 0.679 | 0.565 | 0.565 | 0.542 |
| CNN (ResNet 50) | 0.768 | 0.717 | 0.717 | 0.719 |
| **Our Model** | **0.83** | **0.80** | **0.89** | **0.83** |

## 5.2 DISCUSSION

**Architectural Inductive Bias.** The performance gap between TAI-Speech (AUC 0.839) and the baselines provides empirical validation for the acoustic flow hypothesis. While the Audio Spectrogram Transformer (AUC 0.748) captures global context, its patch-based processing fragments the continuous temporal trajectory required to detect fine-grained motor degradation. Similarly, the static CNN baseline (AUC 0.768) captures spectral texture but misses the velocity of decline. By adapting the Iterative Refinement mechanism from optical flow, TAI-Speech preserves the strict frame-to-frame continuity of the signal, confirming that a strong temporal inductive bias is more diagnostically relevant than global self-attention for non-stationary pathological signals.

**Limitations of Linguistic Pre-training.** The inferior performance of the fine-tuned Wav2Vec 2.0 baseline (AUC 0.679) highlights the risks of using foundation models pre-trained on healthy speech for pathology detection. These models optimize linguistic reconstruction, effectively treating pathological dysfluencies (e.g., slurring, tremors) as noise to be filtered. In contrast, TAI-Speech's ASR-free design explicitly models these mechanical degradations, aligning with the Source-Filter theory where dementia manifests in the instability of the articulatory filter rather than the linguistic source.

**Proximal Biomarkers of Functional Decline.** Although direct IADL scores were not modeled, the high recall (0.89) indicates strong sensitivity to the proximal biomarkers of functional impairment. The Cross-Modal Attention mechanism successfully identifies pathological mismatches such

as silence aligned with rising pitch which serve as interpretable acoustic correlates of the executive dysfunction that precedes downstream IADL failure.

## LIMITATIONS AND FUTURE DIRECTIONS

Despite promising results, this study has several limitations. The findings are based on a constrained dataset from a single linguistic and cultural context, which may limit their generalizability. The cross-sectional nature of the data also precludes any assessment of the model's sensitivity to longitudinal disease progression. Furthermore, the absence of direct IADL measurements restricts the empirical validation of our model's relevance to functional decline. The model's performance on mild cognitive impairment (MCI) also remains an open question for future investigation.

Future work should aim to validate these findings on larger, more diverse, and longitudinal corpora. Incorporating patient IADL scores as an explicit modeling target could provide a more direct method for detecting functional decline. Exploring multimodal fusion, which would combine the temporal acoustic features from TAI-Speech with semantic embeddings from large language models, may also lead to improved robustness and performance. Finally, longitudinal studies are necessary to determine if changes in the model's output correlate with cognitive trajectories over time, potentially enabling the use of personalized baselines for early detection.

## 6   CONCLUSION

We present TAI-SPEECH, a framework that reconceptualizes dementia detection as modeling the continuous dynamics of speech rather than relying on static acoustic categorization. By adapting the iterative refinement mechanism from optical flow, the model captures frame-to-frame acoustic–prosodic evolution without requiring ASR. Our approach surpasses strong baselines including fine-tuned Wav2Vec 2.0, AST, and CNNs, achieving an AUC of 0.83 and a recall of 0.89. These findings indicate that recurrent refinement of acoustic flow provides greater diagnostic sensitivity than methods based on latent linguistic features or global patch-based attention. The ASR-free formulation also mitigates transcription errors and reduces computational and privacy burdens, making it suitable for clinical screening. Our results support acoustic motor instability as a reliable proximal biomarker, consistent with motor-control accounts of cognitive decline. Future work will validate this link using longitudinal datasets with functional impairment (IADL) measures and extend the framework to multilingual speech to assess cross-lingual generalization.

## ACKNOWLEDGMENTS

The authors acknowledge the use of a Large Language Model (LLM) to assist in the editing and refinement of this manuscript. The LLM was utilized to improve grammar, clarity, and conciseness. All conceptual ideas, experimental design, analysis, and conclusions presented in this paper are the original work of the authors.

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
