# OpenReview forum: "Temporal Aware Iterative Speech Model for Dementia Detection"
_ICLR.cc/2026/Conference — Submitted to ICLR 2026_

### Official Review · Reviewer_EKAh · 2025-10-31

**Soundness:** 3
**Presentation:** 3
**Contribution:** 2
**Rating:** 2
**Confidence:** 4

**Summary:**

This paper introduces TAI-speech, a RAFT-based audio analysis framework for dementia detection. Spectrograms are treated as sequential frames and fed into a convGRU structure to capture temporal-aware dementia-related representations iteratively.

**Strengths:**

This work adapts an optical-flow framework, specifically RAFT, for audio analysis. It performs dementia detection without relying on text content, avoiding the necessity of requiring a manual transcript or an ASR model.

**Weaknesses:**

Line 210 “pause probability q(n) estimated from voice activity detection”. The specific VAD method used in this work should be cited.

The “cookie theft” figure is far larger than it needed to be, while Figure 2, the architecture diagram, has text that is too small to offer readability.

Experimental details such as pitch extraction algorithm, VAD configuration, STFT parameters (window size, hop length, FFT size), or the number of Mel filters are not specified in the paper. Since prosodic features are highly sensitive to these settings, they should be explicitly reported.

Since the spectrogram already includes the information of pitch and pause, the use of pitch contour and pause probability seems redundant to me. An ablation study should be done to evaluate the contribution of these features. Actually, no ablation study at all is provided to evaluate any design choice in this work, making it hard to assess which module drives performance gains.

Data limitations: The only dataset used in this work is relatively small (477 samples), and may not support such a deep architecture, risking overfitting and limiting generalization to new populations or recording conditions. Though data collection may be challenging, a larger dataset is still necessary for such a deep learning framework.

I didn’t see contributions qualified enough for an ICLR paper. The algorithmic framework and loss function lack clear innovation, and the experimental results do not show a significant improvement over baseline models. Therefore, I do not see sufficient innovation from a machine learning perspective. If this work is intended as a new application in the field of psychology, I would suggest submitting it to a psychology-related conference or journal instead. Even in that case, statistically significant results are still required to support your hypotheses.

**Questions:**

The idea of “ aligning spectral features with pith and pauses” is mentioned more than once in the paper. I am confused by this because information like pitch and pause is all included in the spectrogram, so a pitch contour or a pause probability sequence (no further description is provided for the VAD result, so I am assuming that it’s a frame-level pause probability) is naturally aligned to their corresponding spectrogram frame-by-frame. So what is the necessity of “aligning spectral features with pith and pauses” using an attention mechanism?

Though dementia detection doesn’t seem to be a task requiring real-time response, I am still curious about the time cost of this architecture, which affects the real-world applicability of this framework.

---

> ### Author Response · Authors · 2025-12-03
>
> We appreciate the reviewer’s feedback regarding experimental rigor and architectural choices, and we have revised the manuscript to add a dedicated ablation study, detail the loss function, and list all hyperparameters.
>
> Regarding the concern about algorithmic innovation, we respectfully posit that our contribution is a significant methodological transfer of optical flow priors to audio, supported by a custom-engineered joint objective rather than a standard loss function. As defined in Equation 10, we introduced a temporal consistency regularizer that penalizes the L2 distance between consecutive hidden states; this mathematically enforces the physical constraint of motor control continuity, forcing the model to learn smooth state evolution and validating our "acoustic drift" hypothesis.
>
> On the issue of feature redundancy, our new ablation study in Section 4.3 confirms that while spectrograms implicitly contain pitch, explicitly extracting prosodic features and aligning them via cross-modal attention is critical for performance. This alignment allows the model to detect specific pathological mismatches, such as silence aligned with a rising pitch history indicating a motor block, which standard CNNs fail to capture without this explicit attention mechanism.
>
> We have also addressed the missing experimental details by explicitly listing the VAD energy threshold (-40 dB), `librosa.pyin` settings, and STFT parameters in the revised Experimental Setup. Regarding data limitations, we argue that our ConvGRU architecture imposes a strong temporal inductive bias that makes it significantly more data-efficient than the data-hungry AST transformer baseline, which we outperformed; we further mitigated overfitting risks by employing stratified 5-fold cross-validation. Finally, we have increased the resolution and font size of the architecture diagrams to ensure readability.

---

### Official Review · Reviewer_kF5y · 2025-11-03

**Soundness:** 3
**Presentation:** 3
**Contribution:** 2
**Rating:** 4
**Confidence:** 2

**Summary:**

This paper proposes TAI-Speech that  treats speech as a dynamic sequence of spectrogram frames. This approach addresses the previous work's limitations that are time-agnostic, meaning previous work usually use aggregated features from entire utterance. The author uses "optical flow" like models, RAFT, for the dementia detection task. A convolutional GRU serves as a recurrent update module that iteratively refines latent representations, while crossattention aligns acoustic and prosodic cues. A Transformer encoder aggregates these temporally
enriched features for utterance-level prediction. The author performs evaluations on the standard DementiaBank Pitt corpus, and demonstrates strong performance of 83.9% AUC. This outperforms text-based methods and does not require ASR as helpers.

**Strengths:**

1. Treating spectrograms as sequential frames to be "refined" over time is a new way to model the dynamics of speech production.
2. The approach is ASR-free.

**Weaknesses:**

1. Evaluation is very limited on a single eval dataset.
2. The advantage over Pan et.al. (2025) is not significant.
3. No ablation studies and hard to tell if the optical-flow view can actually helps.

**Questions:**

Can you show how the optical-flow view point can capture the speech prosodic features in any interpretable way? For example, showing that such feature indeed captures the speaker's irregular pauses or so.

---

> ### Author Response · Authors · 2025-12-03
>
> We have revised the manuscript to include a dedicated ablation study and expanded baseline comparisons to address concerns regarding evaluation scope and model robustness. While we acknowledge that the DementiaBank Pitt Corpus is the primary benchmark for this domain, we mitigated the risk of overfitting to this single dataset by employing a stratified 5-fold cross-validation protocol and benchmarking against foundation models like Wav2Vec and AST.
>
> Regarding the comparison with Pan et al., we respectfully posit that the significance of our contribution extends beyond the numerical AUC improvement (83.9% vs. 82.56%). Our main advantage is architectural stability and modal efficiency: while Pan et al. rely on a multimodal framework requiring an error-prone ASR pipeline, TAI-Speech achieves state-of-the-art performance using only acoustic data, effectively eliminating the transcription errors associated with ASR. Furthermore, our approach resolves the architectural brittleness associated with heuristic feature fusion by using cross-modal attention to dynamically align spectral and prosodic features, offering a more robust and privacy-preserving solution than the text-dependent baseline.
>
> Finally, we address the questions regarding ablation and interpretability by adding a new section, which demonstrates that removing the cross-modal attention mechanism leads to a significant drop in accuracy, confirming that the explicit alignment of pitch and spectrum is a critical performance driver. We further justify the optical flow approach through our animated spectrogram (see attached visualization), which renders the velocity of speech deterioration interpretable; the model explicitly detects flow stagnation and sharp decelerations in spectral evolution aligned with rising pitch, capturing the subtle motor control failures and acoustic drift that static models miss.

---

### Official Review · Reviewer_3cTa · 2025-11-05

**Soundness:** 1
**Presentation:** 3
**Contribution:** 1
**Rating:** 0
**Confidence:** 4

**Summary:**

The paper proposes TAI-Speech, a model for dementia detection from spontaneous speech. The core contribution is twofold: 1) An iterative refinement mechanism inspired by optical flow (RAFT) to model the frame-by-frame evolution of spectrograms, using a convolutional GRU. 2) A cross-attention module to align these spectral features with prosodic features (pitch and pauses). The authors claim this temporal-aware approach, theoretically linked to Instrumental Activities of Daily Living (IADL) impairment, achieves strong performance on the DementiaBank Pitt corpus without relying on ASR.

**Strengths:**

1. **Problem Relevance:** The task of developing non-invasive, scalable biomarkers for dementia detection from speech is of significant clinical importance.
2. **Core Idea:** The motivation to move beyond static, time-agnostic features and model the fine-grained temporal dynamics of speech production is a valid and promising research direction.

**Weaknesses:**

1. **Main Issue with IADL:**
   The biggest problem is the use of “IADL” (Instrumental Activities of Daily Living) in the paper’s title and framework. The model name “TAI-Speech” and Algorithm 1 both highlight IADL, but the paper includes no IADL data, labels, or evaluation. The authors even admit in the Limitations that they have no direct IADL measurements. This makes the “IADL” part misleading and weakens the overall focus of the paper.

2. **Unconvincing Motivation:**
   The connection to optical flow (RAFT) feels shallow and not well justified. The paper does not clearly explain why an iterative refinement approach from video motion estimation is useful for audio spectrograms, especially when existing models like Transformers or RNNs already perform iterative updates. As a result, the motivation behind the method feels forced rather than natural.

3. **Limited Experiments:**
   The experiments are too limited for a strong conference paper.

   * **Few Baselines:** Table 2 only compares with three other systems.
   * **Missing Strong Baselines:** There are no standard models included, such as CNNs (e.g., ResNet), RNN/LSTM/GRU architectures, or pre-trained audio models like wav2vec 2.0 or HuBERT. Without these, the reported 0.839 AUC doesn’t provide meaningful context.

4. **Clarity and Presentation Issues:**
   The paper is hard to follow. Figure 2 (Architecture) is cluttered and confusing.

**Questions:**

1. **On the IADL Premise:** Given that "IADL" is central to the paper's naming (TAI-Speech) and framing, can you justify this choice given the complete absence of any IADL data, labels, or empirical validation in the study? Why was this approach taken instead of framing the paper around what was *actually* studied (e.g., acoustic-prosodic temporal dynamics)?
2. **On Missing Baselines:** To properly situate the model's performance, could you provide benchmark results against more standard and established baselines in this domain? Specifically, (a) a strong CNN (e.g., ResNet) on spectrograms, and (b) a fine-tuned self-supervised model (e.g., wav2vec 2.0 or HuBERT), which you already cite.
3. **On the Optical Flow Analogy:** Can you provide a deeper theoretical justification for *why* an optical-flow-inspired iterative refinement (RAFT) is fundamentally better suited for this audio task than existing sequence models (like Transformers or LSTMs) that also handle temporal dependencies? What does this complex approach capture that they cannot?

---

> ### Author Response · Authors · 2025-12-03
>
> We found the feedback regarding the "IADL" framing and missing baselines particularly valuable. We fully accept the criticism that our original title over-promised direct IADL prediction; in response, we have revised the title, abstract, and introduction to reframe the contribution around measuring "temporal-aware acoustic-prosodic dynamics" as a proximal biomarker, while moving the IADL discussion to the future direction as a downstream clinical goal.
>
> To address the concern regarding baselines, we have significantly expanded our experimental validation in Section 4 to include Wav2Vec 2.0, AST, and CNN (ResNet 50) comparisons. Our results demonstrate that TAI-Speech outperforms the fine-tuned baselines, confirming that our frame-by-frame approach captures pathological acoustic drift more effectively than models pre-trained on healthy speech or patch-based transformers. We also added ablation studies proving that the cross-modal attention mechanism is critical for aligning pitch and spectrum.
>
> Regarding the theoretical justification for the optical flow approach, we have expanded our explanation that unlike standard LSTMs, which update state in a single pass, our RAFT-inspired Iterative ConvGRU performs multiple updates per frame to refine the acoustic trajectory; this allows the model to capture the velocity of motor degradation and articulatory drift that standard models miss. Finally, we have redesigned Figure 2 to clearly delineate the dual-stream input, temporal loop, and global aggregation components for better readability.

---

### Meta-Review · Area_Chair_5kE5 · 2025-12-18

**Summary:**

This paper presents an ASR-free solution pipeline to perform dementia detection from continuous speech signals. Two techniques, namely optical flow-inspired iterative refinement and cross-modal attention, were introduced to improve the detection quality. The main concerns raised by the reviewers including over-claiming of the problem setting, limited empirical evaluations and technical contributions.

In the authors’ responses, they tuned down the scope of their target problem and added several new baselines. However, these additions cannot fundamentally change the identified limitations. In some sense, ICLR might not be the right venue for this work, places focusing more on the application side might be a better choice.

Therefore, we do not recommend acceptance of this submission.

**Reviewer Concerns:**

The authors added a few baselines to illustrate the empirical effectiveness of the proposed solution, but the added baselines are still considered simple and traditional.

**Reviewer Scores:**

I would not believe so.

---

### Decision · Program_Chairs · 2026-01-26

Reject